# The Experiences of a Complex Arts-Based Intervention for Patients with End-Stage Kidney Disease Whilst Receiving Haemodialysis: A Qualitative Process Evaluation

**DOI:** 10.3390/healthcare9101392

**Published:** 2021-10-18

**Authors:** Claire Carswell, Joanne Reid, Ian Walsh, Clare McKeaveney, Helen Noble

**Affiliations:** 1School of Nursing and Midwifery, Queen’s University Belfast, Belfast BT9 7BL, Northern Ireland, UK; j.reid@qub.ac.uk (J.R.); c.mckeaveney@qub.ac.uk (C.M.); helen.noble@qub.ac.uk (H.N.); 2School of Medicine, Dentistry and Biomedical Sciences, Queen’s University Belfast, Belfast BT9 7BL, Northern Ireland, UK; i.walsh@qub.ac.uk

**Keywords:** haemodialysis, arts in health, kidney disease, qualitative, process evaluation

## Abstract

The global prevalence and burden of end-stage kidney disease (ESKD) is increasing, partially as a result of an aging population. Patients with ESKD who receive haemodialysis experience a difficult, protracted treatment regimen that can negatively impact mental health and wellbeing. One way of addressing this issue could be through the provision of arts-based interventions during haemodialysis treatment. A process evaluation was conducted as part of a larger feasibility study, to explore experiences and acceptability of an intra-dialytic (during haemodialysis) arts-based intervention. Thirteen patients and nine healthcare professionals were interviewed. The interviews were transcribed verbatim and thematically analysed. Three themes related to the experience of art on dialysis included: the perception of art participation, as patients described initial anxiety around participating in art, which reduced after they experienced the intervention; the benefits of art participation on both patients and healthcare professionals; the benefits including improved self-esteem, motivation, increased social interaction and an overall improved haemodialysis experience; and the acceptability of the arts-based intervention, as the intervention was highly acceptable, even when delivered by a facilitator who was not a professional artist. This study highlights that arts-based interventions could be used to improve the mental health and wellbeing of patients with ESKD receiving haemodialysis.

## 1. Introduction

End-stage kidney disease (ESKD) is the final stage of chronic kidney disease (CKD) and is reached when a person’s kidneys can no longer adequately filter their blood, resulting in kidney failure. ESKD is associated with difficult physical symptoms and an intensive treatment regimen [1]. Treatment for ESKD typically involves some form of renal replacement therapy (RRT), such as dialysis or kidney transplantation. Kidney transplantation is the gold standard treatment for ESKD [1]; however, this option is restricted by the availability of viable kidney donors and the risks associated with surgery in older, frail patients [2]. The most common form of dialysis is in-hospital haemodialysis, where patients attend hospital three times a week to receive haemodialysis treatment over a period of four hours [3]. As haemodialysis is a life-sustaining treatment, most patients require this for a prolonged period of time until they receive a kidney transplant and, in circumstances where patients are not eligible for kidney transplantation, they will need to receive haemodialysis for the rest of their life.

This treatment has holistic implications for health and wellbeing as it can negatively impact both physical and mental health, leading to an overall reduced quality of life. One component of the treatment experience that is thought to contribute to this negative impact on mental health and wellbeing is the issue of empty time during haemodialysis [4]. Patients have limited opportunities to engage in meaningful activities during their haemodialysis treatment and this creates profound boredom resulting in rumination and contemplation of illness and death [5]. Arts-based interventions could be a potential vehicle to address this issue as they can provide a meaningful, engaging activity during a difficult treatment. 

Additionally, arts-based interventions have demonstrated some efficacy in reducing symptoms of anxiety and depression in other patient populations [6]. The improvement in these symptoms could be achieved by inducing a ‘flow state’, a psychological phenomenon characterised by complete absorption in an activity, the experience of time passing quickly (tachypsychia), a reduction in self-consciousness, rumination and worry [7,8]. However, there is a lack of evidence assessing their efficacy for improving health and wellbeing in patients receiving haemodialysis. The existing evidence base relies predominantly on observational studies [9] or programme evaluations [10], which suggest that these interventions may be beneficial for the mental health and quality of life of patients with ESKD receiving haemodialysis [8]. The lack of randomised controlled trials (RCTs) of complex arts-based interventions for people with chronic illnesses is an established issue [11] and could result from difficulties surrounding implementation of complex arts-based interventions in clinical areas, manualising arts-based interventions in a replicable way, lack of acceptability of complex arts-based interventions, dependency on facilitator factors and lack of appropriate outcome measures. Therefore, this study aims to explore the acceptability of the delivery of an arts-based intervention, as part of a pilot cluster RCT, in a haemodialysis unit.

## 2. Materials and Methods

### 2.1. Design

Qualitative process evaluation. The process evaluation was conducted in parallel to a pilot cluster randomised controlled trial (RCT) that aimed to evaluate the feasibility of evaluating a complex arts-based intervention within the framework of an RCT [12,13].

### 2.2. Participants

Participants included both patients who had been recruited into a pilot cluster RCT of a complex arts-based intervention [12] and healthcare professionals working on the haemodialysis unit in the UK during the implementation of the intervention. The mean age of participants in the pilot cluster RCT was 69 years, while the mean years participants had been receiving haemodialysis was 4.3 years. The protocol for the pilot cluster RCT [13] and results of the pilot cluster RCT [12] have been published in detail previously. All participants in the pilot cluster RCT who started the intervention completed the full six sessions of the intervention.

Eligibility criteria for patients:Age 18 or over;Able and willing to participate, as assessed by a member of the nursing staff;Receiving haemodialysis;Recruited into the pilot cluster RCT of the arts-based intervention.

Eligibility criteria for healthcare professionals: A member of the multidisciplinary team, including nurses, healthcare support workers, doctors, dietitians, social workers and counsellors;Have had experience with the intervention, meaning they had been present on the unit during implementation of at least one session of the intervention;Have worked in a clinical renal setting for more than three months. Familiarity with the context of the clinical environment was needed to inform the acceptability of the intervention [14].

### 2.3. Recruitment

A purposive sampling strategy was used. Patients who had been recruited into a pilot cluster RCT of the arts-based intervention were offered the opportunity to participate in the process evaluation following delivery of the intervention. Healthcare professionals were recruited for the process evaluation according to the proportion of time they spent on the unit during delivery of the intervention. The ward manager acted as a gatekeeper and screened healthcare professionals to ensure that they met the inclusion criteria. Informed consent was then collected at the start of each interview. A total of 13 patients consented to semi-structured interviews and the interviews continued until data saturation was reached [15]. The sample included nine participants from the experimental group, including five men and four women, and four participants from the wait-list control group, with two men and two women, of the pilot cluster RCT. This ensured exploration of both experiences and expectations of the arts-based intervention. A total of nine semi-structured interviews were completed with healthcare professionals before data saturation was reached. The healthcare professionals who were interviewed included a ward manager, six band 5 nurses and two healthcare assistants.

### 2.4. Data Collection and Management

The semi-structured interviews consisted of open questions informed by the RE-AIM QuEST framework [16]. The RE-AIM framework outlines that the reach, effectiveness, adoption, implementation and maintenance of an intervention should be explored [16]. The interview schedules are available as Appendix A.

The location of the interview was dependent on participant preference. The majority of patients were interviewed whilst receiving haemodialysis, while one patient was interviewed in a private room on the unit prior to their haemodialysis session. Healthcare professionals were interviewed in the haemodialysis unit in a private office. The interviews with healthcare professionals and patients from the control group were conducted by the lead author, while the interviews with the patients who received the intervention were conducted by HN and CMcK.

### 2.5. Data Analysis

The semi-structured interviews were recorded and transcribed verbatim and thematically analysed using NVivo Version 11 [17]. Inductive thematic analysis [18] was used to analyse the data collected. The data was divided into three distinct data sets: interviews from participants in the experimental group, interviews from participants in the control group and interviews with healthcare professionals. The first step involved descriptive coding of each data set, line by line. During the second step, interpretive codes were synthesised from the descriptive codes to provide a preliminary description of the data. The data sets were then merged together to arrange the codes from all three data sets into hierarchical categories, forming final overarching themes that captured the qualitative data as a whole [19]. Investigator triangulation was used to ensure validity of the identified themes. This involved regular review of the analysis and themes by the research team to develop consensus on the final identified themes [20].

### 2.6. Intervention

The intervention that was delivered as part of the pilot cluster RCT involved six one-hour long one-to-one facilitated art sessions during haemodialysis, provided twice a week over a period of three weeks. The intervention was developed in collaboration with patients and healthcare professionals [21] and was informed by the theory of Flow [7]. During the development of the arts-based intervention it became important to distinguish the arts-based intervention from art therapy. This intervention is not considered ‘art therapy’ as it does not follow a formal psychotherapeutic framework and was not delivered by a trained art therapist [8].

The facilitator was a trained mental health nurse, researcher and an amateur artist with no professional or higher education qualification in the arts. During each session, participants could choose between visual arts and creative writing, and could choose between different types of visual art (graphite pencil, coloured pencils, water colour painting and fine liner pens) and different types of creative writing (poetry, short stories and storytelling). The sessions were intended to be person-centred and flexible but underpinned by a focus on skill development. In order to induce flow state, people need to be engaged in an activity that they find challenging but in which they develop the skills to meet that challenge [21].

## 3. Results

The themes and subthemes are presented in Table 1.

### 3.1. Theme 1—Perception of Art Participation

This theme highlights the interviewee’s perception of art participation prior to the intervention, during delivery and following completion of the intervention. Two subthemes highlight the changed perception of art participation over time, describing preconceptions and the perception of art after the intervention. The third subtheme describes the negative appraisal of artistic abilities by participants, which was consistent throughout the study.

#### 3.1.1. Mixed Preconceptions and Apprehensions of Art Participation

Patients and healthcare professionals had limited prior experience of participating in the arts, with most patients stating their most recent engagement had been at primary school. As a consequence, participants had limited knowledge surrounding what to expect, but the novelty of the activity was a motivating factor to participate:

*Just look forward to something different, that’s me*.—CG03

Healthcare professionals had a similar lack of experience. In some cases, this instilled a sense of curiosity and interest. Even without any prior experience of arts within a healthcare setting, the majority of healthcare professionals were positive and open to delivery of an arts-based intervention within the haemodialysis unit:

*I was thinking ‘oh I wonder how this is going to go and how’s it going to work’. Yeah, very positive. I thought it was a great idea*.—HCP05

The one consistent expectation patients reported, which acted as a motivating factor for participation, was that arts activities would occupy their time on dialysis. The opportunity to engage in a new activity to “*try and do something*” was appealing to most participants:

*Never was interested in art but I thought I would give it a try and it might help me, you know, in future. Whenever I’m lying here, you know*.—EG08

Despite these motivating factors, participants and healthcare professionals had concerns prior to delivery of the intervention. The primary concern patients related to their lack of artistic abilities and skills. They anticipated that they would be unable to produce artwork of adequate quality due to a perceived lack of talent:

*And I thought I’d just make a complete mess*.—EG06

Healthcare professionals reported concerns related to the practicalities of patients engaging in the arts within the dialysis setting, both in relation to the clinical setting, patient health and their level of engagement. Restrictions of the clinical area, for example, the constraints of dialysing through an arteriovenous fistula that restricted the use of an arm alongside the clinical duties of the healthcare professionals, were identified as potential barriers to implementation:

*My first thought of it [the arts-based intervention] was like how would this even be possible with some patients with only one arm, um, able to do stuff*.—HCP01

#### 3.1.2. Improved Perception of Art Participation Following Intervention

While participants entered the study with a degree of anxiety, they reported change in their perception of art participation following delivery of the intervention. The changes in perception were positive, with participants identifying an interest where they had none previously: 

*Yes, it would have changed from me initially… From going from, uh, not being interested. Sort of thing, at the start I was just doing it cause it was something to do, till enjoying it now*.—EG03

This resulted in a desire to engage in the arts in the future. Patients were provided with their own arts pack containing a sketch book and materials for drawing, painting and writing whilst on haemodialysis. Each participant kept this pack at the end of the study. The arts pack also acted as motivation for participants to continue with the arts as it provided the necessary materials and acted as an external prompt.

*When I seen the wee bag from Queen’s University sitting over in the thing it always came back into my head, here’s me ‘well I must take that out tomorrow and try and do something’*.—EG04

#### 3.1.3. Negative Appraisal of Own Artistic Abilities

While participants perceived art as an enjoyable activity after experiencing the intervention, they had a consistent negative appraisal of their own artwork and abilities. While participants acknowledged they lacked experience, they were quick to disparage their initial work:

*The first one was absolutely dreadful! Because it was painting, and I just ended up with this… blobs all over the place. It was supposed to be a scene of you know, eh, by the beach, a beach and water and clouds. Awful! Blob. Blob. Blob*.—EG06

Participants’ views on their own artwork improved over time as their skills improved, but a highly critical view of their own work continued: 

*Well, I don’t think that any of it’s perfect, but it got better*.—EG03

### 3.2. Theme 2—Benefits of the Arts-Based Intervention on Patients and Healthcare Professionals

This theme highlights the different beneficial experiences related to the arts-based intervention. It includes five subthemes that describe the benefits described by participants and healthcare professionals. While these benefits are presented as distinct subthemes, they likely interact to provide a holistic effect.

#### 3.2.1. Generation of Positive Affect Due to Exposure to the Arts-Based Intervention

Patients stated that the arts-based intervention generated a positive affective response. Patients reported that participating in the intervention made them feel happy and that the main benefit they experienced was enjoyment, whilst healthcare professionals also noticed this positive change in affect.

*I think they really enjoyed it, like, you know, they definitely—it was great, I would love to see it in the unit. You know the patients… you get a different side to the patients, and they were so much happier, I think*.—HCP09

In addition, patients recounted how the arts-based intervention helped them relax. In contrast, other participants reported that art was stimulating and helped them remain awake whilst receiving dialysis. The underlying mechanism was the ability of art to occupy a person’s mind whilst receiving treatment. This prevented them from attending to worries or concerns, allowing them to relax.

*Very relaxed and away from me—well, not leaving all my worries behind but just sort of, made me a lot more relaxed than what I would be*.—EG07

Healthcare professionals also described an impact of the intervention on their own affective states. In particular, they described how experiencing the patient’s outward enjoyment of the intervention impacted their mood.

*As in we were aware that you were going round and doing things and we could see the patients engaging, for me that was a nice thing to be able to see*.—HCP05

#### 3.2.2. Improved Self-Esteem

Participants were able to recognise the skills they had developed and their improvement throughout the intervention. As a consequence, they experienced a sense of pride and accomplishment:

*Oh, well, I suppose satisfaction that you can add another string to your bow at 80 odd years of age*.—EG03

This sense of achievement came from participants’ realisation that they were capable of something they had previously thought they were unable to do. Participants were able to recount how the facilitated sessions introduced them to principles and techniques that they had previously been unaware of. The ability to produce artwork provided tangible evidence that they had the capacity to learn and develop new skills: 


*Just doing it was enough to let you see that you were capable of doing something, you know?*
—EG01

This improvement in self-esteem was also described by healthcare professionals. They observed the change in mindset participants had in terms of their capabilities:

*A good thing because it gave them the understanding of maybe... what em… not what they were expected to do but maybe of what they could do themselves. What they were capable of*.—HCP02

#### 3.2.3. Sense of Purpose

The arts-based intervention provided participants with an external motivation. Participants experienced a sense of purpose throughout the intervention’s delivery as it gave them an activity to work towards and a skill to develop:


*Well, it gave me something to go for, like a goal… gave me a goal to go for, you know… You go for that and you keep going for it, you know. And before you know it it’s over and done with, you know?*
—EG02

As a consequence of this sense of purpose, participants started looking forward to attending the dialysis sessions. This seemed to have a particular benefit within this clinical setting due to the lack of meaningful activity that participants engage in during haemodialysis.

*It was a good wee experience, gave me something to do, and I more looked forward to coming in to dialysis because I knew I had something to do when I came in. I wasn’t just going to lie there and start thinking again*.—EG04

This increase in motivation and excitement amongst participants was observed by healthcare professionals. They noticed a change in patients before their dialysis sessions, as they appeared more motivated to come to dialysis and expressed excitement about the intervention.

*Well you could see a difference in the patients even out in the reception by them talking about ‘oh what did you do the last day, what are you going to do today?’ and you could see the excitement in them. Em, and it got the patients more … they were looking forward nearly to come to see what they were going to do and see what they were going to learn*.—HCP04

Participants reported thinking about their art in between sessions in order to identify subject matter they found interesting. Healthcare professionals thought this had a further benefit as the sense of purpose extended outside the clinical environment and into their daily lives:

*You know it made them sort of more active at home as well, to think ‘right, okay, what could I… what’s of interest to me that I could actually take into Claire [the facilitator] and draw’ so… I think that’s good aspects at home and within the unit*.—HCP01

#### 3.2.4. Increased Social Interaction

The arts-based intervention was a collaborative and interactive intervention that required consistent communication with the facilitator. Consequently, there was a large social component to the intervention. Participants identified this social aspect, the fact they had company during dialysis sessions, as highly beneficial:

*There was somebody there to talk to. The talking to was just as good as the art… Well it’s just I loved the company of [the facilitator]; the company made a great difference*.—EG04

Healthcare professionals contrasted the role of the facilitator with the role of the healthcare professional, suggesting that the relationship between the facilitator and participant had a positive impact as the focus was the creation of art, as opposed to a clinical relationship centred on illness: 

*So, they were getting that one-to-one time where it was something that was about them and it was positive rather than them being sick and me standing over them. So, I thought it was great*.—HCP05

Healthcare professionals also identified benefits to social relationships in the unit as the intervention facilitated communication between patients and healthcare professionals. This was achieved through a common interest amongst patients and a non-clinical focus of conversation between patients and healthcare professionals:

*So, you could see patients who ordinarily wouldn’t have talked, em, beginning to communicate with each other. So that was nice to see, and it became nearly like a talking point of like the patients… who was participating in it and how they were getting on*.—HCP04

#### 3.2.5. Positive Influence of the Arts-Based Intervention on the Dialysis Experience

There were a number of ways through which the art had a beneficial influence on patients’ experiences of receiving haemodialysis. Distraction was identified as a positive part of the intervention. The two mechanisms through which distraction was beneficial was through the avoidance of difficult thoughts, either concerns or ruminations, and distraction from the clinical treatment and disease itself: 

*You were doing something with your hands, you were focusing on trying to do this well… Because I was doing something instead of just lying thinking about different things. Your mind just goes, you go from one thing to another type of thing, you know thinking back to the past*.—EG04

Related to the benefits of distraction was how this influenced the patients’ perceptions of time during haemodialysis, specifically by making the time pass quicker:

*Aye the time seemed to go in a bit quicker whenever … comes in, you watch TV but uh, you never notice the time so much. But whenever you’re painting time, that hour or whatever it was that she was in with, just seemed to go like that*.—EG05

The influence of the intervention on the overall dialysis experience also benefited healthcare professionals. Healthcare professionals reported that the intervention changed the environment, transforming the atmosphere into one where the focus was not exclusively clinical: 

*It made it look like we just didn’t see it as a dialysis environment, that we were actually looking at different aspects of the patient’s wellbeing*.—HCP04

### 3.3. Theme 3. Acceptability of the Arts-Based Intervention

This theme describes the acceptability of the arts-based intervention. There are five subthemes that relate to the intervention’s acceptability, addressing the intervention, implementation strategies and the context in which the intervention was delivered.

#### 3.3.1. Adaptation of the Arts-Based Intervention to the Constraints of a Haemodialysis Setting

Whilst the intervention was developed to ensure ease of implementation in this context [21], there were still a number of barriers to delivery that were identified by participants. One of the main concerns was that vascular access required for haemodialysis frequently meant patients only had a single hand available for the activities. For one participant, in particular, this meant they had to use their non-dominant hand to paint:

*See my hand’s tied down. I wouldn’t be able to paint with my right hand, that means I had to paint with my left hand*.—EG01

Patients receiving haemodialysis also require continuous monitoring of their blood pressure, which produces an additional barrier to delivery as the blood pressure cuff is placed on the arm that is not being used for vascular access. This resulted in patients having to take small breaks during each session when the cuff inflated, as movement of the arm can interfere with the blood pressure reading:

*I’m right handed and then I have to have the cuff on me right hand and then you’re doing something—you’re writing or drawing—and then your blood pressures being taken and you have to sit and you know, leave it till it finishes but you can’t—you need your blood pressure take quite often*.—EG02

Participants also identified mediating factors that allowed these barriers to be overcome. Preplanning and flexibility from the facilitator were crucial; the materials had been pre-selected for ease of use, there was a selection of materials to cater to people with different abilities and the timing of the sessions could be changed if needed:

*Not even patients with fistulas! Did that affect them? No! (Laughs) Not at all! So, if you can get over that barrier and they’re happy—yeah, there were no problems*.—HCP09

#### 3.3.2. Positive Influence of Facilitator on Participant Experience

The importance of facilitation was described by both patients who received the intervention and healthcare professionals. Whilst a positive relationship with the facilitator promoted engagement, the relationship was also highly collaborative and ensured that participants were comfortable with, and interested in, the activities on offer: 

*I made her draw everything with me (laughs) but her and me got on the best so it was real interesting like, so it was, I liked it… I says you need to draw it along with me so. And that made me feel good too—she was a good artist too*.—EG05

Another aspect of the facilitation that promoted engagement was consistent encouragement and positive feedback. Healthcare professionals identified the importance of encouragement during facilitation to reduce anxiety participants may have about the intervention.

*She says ‘Well try [participants name]’ She says ‘it’ll be your own painting, it’ll not be the same, nobody’s two paintings are the same. ‘Cause that’s your painting, and that other one’s somebody else’s, what they say, just paint what you see… Yes, it made me feel good because she thought it was good and she was learning me*.—EG04

One-to-one facilitation, as opposed to group, was essential for delivery. Healthcare professionals specified this was important for the dialysis setting due to the restrictions of the treatment, but also for the benefit of the patients, as they felt that the intervention had to be catered to the individual to ensure participants were comfortable:

*One-to-one is very good! Which meant, you know that everybody was an individual, like these art things—especially people here, if you sit them in group, maybe everyone doesn’t want to discuss their problems together*.—HCP06

#### 3.3.3. Importance of Participant Choice on Subject and Activity within the Arts-Based Intervention

Participants who engaged in the intervention recounted how individual choice of subject matter and activity influenced the work they created and were able to identify artistic pieces they produced during the intervention that captured their own interests and experiences:

*She said ‘what would you like to do next time?’ So, I said ‘what about father Christmas?’ So, I had a little tiny novelty father Christmas stuck on my fridge and I brought it in. And he had these long dangly legs, and a funny wee face. So, I managed to do that*.—ECG06

Variety maintained participant interest and allowed participants to experiment and identify what activities they preferred. Personal preferences informed the selection and contributed to a positive experience:

*She [the facilitator] started on the writing and poetry and telling wee stories and I love that, I used to belong to a wee writing class and that so*.—EG07

The importance of this choice in participant engagement was highlighted by healthcare professionals, who also attributed the variety of choice in the subject matter, artistic activity and the materials as an important factor that aided delivery:

*Because they weren’t pigeonholed in that ‘you have to do this, you have to do that’ it was all their input, what they wanted to do. And you seemed to have a magic bag of tricks that ‘Yeah, I’ve got that!’ It was great! But as for the clinical area it didn’t impact whatsoever, and you obviously had it well thought out that you had everything available for them which was great*.—HCP05

#### 3.3.4. Length of the Arts-Based Intervention

The one consistent criticism of the intervention provided by both participants who engaged with the intervention and healthcare professionals, was how long the intervention lasted, both in terms of the length of the sessions and the number of sessions provided. Some participants suggested the sessions themselves should be longer due to the amount of time that they spend on haemodialysis:

*But some days I thought it was pretty short and other days… well it wasn’t too long, it was never too long*.—EG02

Participants who received the intervention and healthcare professionals consistently stated that the intervention, as a whole, needed to be longer in terms of more sessions being provided over more weeks:

*I’d have liked it to last a bit longer and I’d have liked it to- maybe a bit more time. It just finished too soon*.—EG07

The additional observation that participants who received the intervention required numerous sessions before they felt that they could fully engage with the arts also suggests more sessions are required:

*Awk it was… a bit… not depressed, cause I’m not a depressive person, but I was sad that it was finished*.—EG04

Participants expressed a desire to do more art, yet most did not do so independently following completion of the intervention. The primary reason for this lack of engagement was the absence of the facilitator as there was no longer a person present to provide consistent support and encouragement. Participants also reported that they did not feel they had the confidence or adequate skill set to continue without guidance and felt the quality of their art suffered in the absence of facilitation: 

*Well I just, I accepted it, you know that, that they were finished you know. But I didn’t do anything since I was finished, I never painted anymore*.—EG01

#### 3.3.5. Quality and Suitability of Materials for the Arts-Based Intervention

Each participant received an arts pack that contained a variety of art materials that could be used during the session. There was a consistently positive reaction to the materials that had been provided, with participants highlighting both the quality of the materials and their ease of use:

*They were brilliant to work with, the pencils and all the rest of it, and the paint I liked them as well*.—EG02

Healthcare professionals also highlighted the importance of the choice of materials for use within the clinical context and acknowledged there would be certain arts activities that would not have been feasible or safe to use in a haemodialysis unit:

*It was all… catered to the environment. It was fit for the environment. Even if you used water based, they were not like messy water based… what children would use, you know. So, it was all very… very accurate. Very well done*.—HP06

## 4. Discussion

While this study does not conclusively establish an effect of an arts-based intervention on patients receiving haemodialysis, it does assist in identifying some mechanisms through which the arts-based intervention might improve patient outcomes. The subthemes identified in the process evaluation relating to the benefits of the arts-based intervention included generation of positive affect due to exposure to the arts-based intervention, improved self-esteem, sense of purpose, increased social interaction and the positive impact of the arts-based intervention on the dialysis experience. The identified subthemes suggest that arts-based interventions may promote aspects of wellbeing as they align with constructs of positive psychology, in particular, the elements of wellbeing outlined in the ‘PERMA’ model [22]. The elements identified in this model include positive affect (feeling happy), engagement (opportunities to experience flow states), relationships (feeling integrated into a society or community), meaning (a sense of purpose in life) and accomplishment (making progress towards goals).

The theoretical framework that guided the development of the intervention was ‘flow’ [7], and there is evidence from the process evaluation that flow states were induced during the intervention. For example, the positive impacts on the dialysis experience included distraction from both surroundings, worries and ruminations, and the phenomenon of tachypsychia, as patients experienced time passing quicker on haemodialysis whilst engaged in the intervention. This is consistent with previous qualitative research on arts-in-medicine programmes implemented during haemodialysis, where the primary benefits appear to result from the inducement of flow states [23]. However, the process evaluation identified benefits that suggest the arts-based intervention has an effect beyond the inducement of flow states, such as the development of a sense of purpose, improved self-esteem, happiness and increased social interaction, and that the positive impact of an arts-based intervention may result from a more holistic improvement in wellbeing. These experiences map onto the positive psychological theory of ‘PERMA’ [22].

The intervention also had a positive impact on the experience of haemodialysis, both for patients who were participating in the intervention and healthcare professionals. Whilst the altered perception of the passage of time and distraction from the clinical procedures and setting helped improve the treatment experience, the intervention also transformed the environment of the haemodialysis setting by providing a more humanistic and less clinical focus for both the patients and healthcare professionals. As aesthetic deprivation is an issue in most clinical settings, and is thought to negatively impact wellbeing for both patients and healthcare professionals [24], providing aesthetic experiences and expression in the clinical setting can potentially improve wellbeing, even for those not actively engaged in the intervention. Healthcare professionals also reported that their communication with patients improved as they were provided with a point of conversation that was not related to the patient’s disease or clinical treatments. This more humanistic focus allowed healthcare professionals to feel they were providing genuine holistic care, and this may have implications for the scope of impact that should be evaluated within future trials of arts-based interventions. Environment and organisational structures are known to influence healthcare professionals’ wellbeing and their subjective capacity for providing compassionate care [25]; therefore, an arts-based intervention may be able to improve not only patients’ wellbeing but the wellbeing of staff working in the clinical environment.

Another key finding from the study was the acceptability of the delivery of an arts-based intervention by a facilitator who was not a professional artist. While the facilitator had an interest in arts, they had no higher education relating to art, a contrast to the usual practice of professional artists-in-residence who provide similar activities in hospital settings [23,26]. This finding has implications for clinical practice. It highlights that patients can enthusiastically engage, and experience numerous benefits, with a facilitator experience if the art is restricted to personal interest and practice, potentially influencing costs and sustainability by introducing opportunities for non-professional volunteer involvement [27].

## 5. Strengths and Limitations

This study was conducted as part of a pilot cluster RCT as a means of evaluating the impact of an arts-based intervention on patients with ESKD whilst receiving haemodialysis. One main strength of this study was the provision of consistent guidance from an interdisciplinary advisory group that consisted of healthcare professionals, patients, artists and academics. This ensured that, during implementation, all aspects of the clinical setting had been considered and the intervention was able to fit into the clinical routine of the unit without disturbing healthcare professionals’ duties or being obstructive or burdensome to patients. A limitation of this study was the dual role of the facilitator as both the facilitator of the intervention and lead researcher. However, steps were taken to prevent this from influencing the process evaluation, and the semi-structured interviews were conducted by other researchers with experience in qualitative research. The study also took place at a single site that had a demographically homogenous population; therefore, this study has not been able to identify issues of acceptability that may occur across multiple sites or across demographically diverse locations. Finally, the exclusively positive appraisal of the intervention may have resulted from the relatively small sample of participants who completed the full intervention or the influence of the facilitator on the intervention experience.

## 6. Conclusions

In conclusion, an arts-based intervention for patients receiving haemodialysis is highly acceptable and may provide benefits to the mental wellbeing of both patients and healthcare professionals. Future research should focus on establishing the efficacy of complex arts-based interventions using rigorous trial methodology and developing other holistic interventions to address the mental health and wellbeing of patients with kidney disease.

## Figures and Tables

**Table 1 healthcare-09-01392-t001:** Process evaluation themes and subthemes.

Theme	Subthemes
Perception of art participation	Mixed preconceptions and apprehensions of art participation
	Improved perception of art participation following intervention
	Negative appraisal of abilities
Benefits of the arts-based intervention on patients and healthcare professionals	Generation of positive affect due to exposure to the arts-based intervention
	Improved self-esteem
	Sense of purpose
	Increased social interaction
	Positive influence of the arts-based intervention of the dialysis experience
Acceptability of the arts-based intervention	Adaptation of the arts-based intervention to the constraints of a haemodialysis setting
	Positive influence of facilitator on participant experience
	Importance of participant choice on subject and activity within the arts-based intervention
	Length of the arts-based intervention
	Quality and suitability of materials for the arts-based intervention

## Data Availability

The data set for this qualitative study can be provided by contacting the authors of the paper.

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
