# Peer review of "The Experiences of a Complex Arts-Based Intervention for Patients with End-Stage Kidney Disease Whilst Receiving Haemodialysis: A Qualitative Process Evaluation"

_healthcare, 2021, doi:10.3390/healthcare9101392_

Round 1

Reviewer 1 Report

I think the research is well designed and explained, the references are correct.

However, the article is somewhat repetitive in some sections due to repeated insistence on issues already mentioned such as the duration of hemodialysis or the lack of artistic skills of the participants, for example.

I think some reiterations could be eliminated and add more specific information in everything related to the type of artistic activities that were carried out. A brief reference is made to writing or drawing, but what were the options that patients had? What did they mostly decide to do? Were there the same answers regardless of the artistic activity? If they wrote: did the text have any relation to what they were experiencing?

I miss any more explanation about the type of art, activities, and if their typology had any influence on the results.

I have also missed more information about the facilitator: was he an artist, a healthcare professional or a researcher? Was it just one person or was it several?

And, finally, when the intervention was finished, it is explained that the patients did not continue making art, but did their mood get worse? Those items in which the intervention helped: how did the absence affect them? Did the patients' mood get worse?

Author Response

Thank you so much for taking the time to review our manuscript, for your comments and constructive feedback. It’s greatly appreciated.

However, the article is somewhat repetitive in some sections due to repeated insistence on issues already mentioned such as the duration of hemodialysis or the lack of artistic skills of the participants, for example. I think some reiterations could be eliminated and add more specific information in everything related to the type of artistic activities that were carried out. A brief reference is made to writing or drawing, but what were the options that patients had? What did they mostly decide to do? Were there the same answers regardless of the artistic activity? If they wrote: did the text have any relation to what they were experiencing?

I miss any more explanation about the type of art, activities, and if their typology had any influence on the results.

I have also missed more information about the facilitator: was he an artist, a healthcare professional or a researcher? Was it just one person or was it several?

We have gone through the results sections and removed the repeated references to the participant’s perceived lack of skills and some of the other repetitions that were clear from that section.

We have added in a paragraph detailing the intervention, this should give an idea of the focus of the sessions. The arts-based intervention was not art therapy, meaning that the participants did not have to base the art around their own mental health or emotions. Participants who did engage in creative writing could choose their own prompts or stories that they wanted to tell, while this obviously related to personal experience or related to subjects of great personal significance, the focus was not on content or output but on the process of learning the skill of creative writing or drawing. The distinction between art therapy and arts-based intervention is now briefly described in the intervention section.

The fact that there was only one facilitator has now been highlighted as a limitation in the discussion. We have also provided a brief description of the facilitator in the section about the intervention.

And, finally, when the intervention was finished, it is explained that the patients did not continue making art, but did their mood get worse? Those items in which the intervention helped: how did the absence affect them? Did the patients' mood get worse?

As this was a pilot study and the interviews were only conducted at one point in time during that study, so whether the intervention improved their mood over time could not be explored qualitatively. The quantitative aspects have previously been reported in a separate publication, however as it is a pilot study they should not be interpreted as demonstrating cause and effect, just the feasibility of collecting data over a period of time

Thank you again for your comments!

Reviewer 2 Report

Overall I find this to be a good paper on an important topic (the use of the arts in healthcare) but there are issues that need to be addressed.

Abstract: The three themes that emerged from the qualitative analysis seem overly general and predictable. If possible, more specificity is needed so that the most important aspects of the findings can be easily grasped in the Abstract. "Intra-dialytic" needs to be defined.

Introduction and Rationale: The rationale for the study is built on only a few previous studies. The authors could have used more findings from related research but this a minor concern. They are correct that there have been very few randomized controlled trials about this kind of intervention. They could mention that one of the reasons for the small number of RCTs is that arts-based interventions are difficult to standardize because they are often customized to each patient. Another is that it can be hard to factor out therapist effects, as can be seen in this report.

For a qualitative study like this one, the Introduction and literature review should lead to one or more 'Research Questions' that are stated at the end of the Introduction. These are 'how' and/or 'why' questions rather than 'what' or 'whether' questions. In turn, the Research Questions determine the choice of methodology.

The greatest shortcoming of this paper is that intervention is not described! It needs to be described in detail, right after the section on recruitment. We need to know what the patients experienced: what did they do, how many sessions did they have, how long were the sessions...? How was the intervention derived? Does it mirror interventions used previously in similar settings? Information is needed about the person who provided the intervention. Did she have training or previous experience with this kind of intervention? Does she have art therapy training (North American art therapists reading the paper will want to know this)? This information is important for understanding how the intervention might be used in the future, in other settings.

Some people in the art therapy world might argue that art therapy training is necessary for delivering this kind of intervention. I don't feel that way myself. Rather, I am interested in finding ways for health professionals who are not art therapists to deliver arts-based interventions so arts-based interventions can be more widely available. Toward this end it is important to discuss the skills/credentials/experience of the facilitator and perhaps make suggestions about the qualifications needed.

If any patients dropped out, this needs to mentioned, with reasons given.

The intervention included creative expression (painting, drawing and writing?) and one-on-one facilitation. These two components of the intervention might have been equally important, or one might have been more important than the other. If the authors have data that could shed light on this question it would be helpful to add that information in.

Was it a wait-list control group?

Results: these are well presented, but missing is a discussion of comments that did not fit with the themes. There probably were some things that didn't go well, or some patients who didn't want to continue, or something along these lines. Negative comments, or exceptions to the main themes, need to be mentioned for the presentation of the results to be complete.

Discussion: A qualitative study cannot "conclusively establish an effect" and maybe the words "effect" and "impact" are used too much in the Discussion. Effects and impacts were mentioned by the interviewees, but they were not demonstrated, established or proven. The focus needs to stay on the potential mechanisms that the qualitative analysis revealed.

The Discussion overall is very good. 

A big problem with this kind of study (novel arts-based intervention with only one facilitator) is that it is impossible for readers to assess the facilitator effect. Maybe the facilitator was so nice to be with that anything she did with the patients would have lead to comments about beneficial effects! Maybe she was an outstanding facilitator and a similar intervention delivered by a different person would not have seemed acceptable to patients. This problem needs to be addressed and included as a limitation.

A larger study on this topic, whether an RCT or some other type of study, would need to have more than one facilitator. 

One of the reasons I think this paper is important is that arts-based interventions can have powerful positive effects in healthcare settings at relatively low cost. The main question in my mind after reading the paper was: who can facilitate this intervention for ESKD patients? Can it be a nurse who is there anyway, perhaps a nurse with some experience with the arts? Does it need to be someone else? If so, who pays for that? Etc.

Congratulations on an excellent paper.

Author Response

Thank you so much for taking the time to review this manuscript, and for providing so insightful and constructive comments.

Overall I find this to be a good paper on an important topic (the use of the arts in healthcare) but there are issues that need to be addressed.

Abstract: The three themes that emerged from the qualitative analysis seem overly general and predictable. If possible, more specificity is needed so that the most important aspects of the findings can be easily grasped in the Abstract. "Intra-dialytic" needs to be defined.

The themes have been reframed slightly, and additional specificity given to the final theme which may be particularly important for the implications on practice.

The term ‘Intra-dialytic’ has now been defined in the abstract.

Introduction and Rationale: The rationale for the study is built on only a few previous studies. The authors could have used more findings from related research but this a minor concern. They are correct that there have been very few randomized controlled trials about this kind of intervention. They could mention that one of the reasons for the small number of RCTs is that arts-based interventions are difficult to standardize because they are often customized to each patient. Another is that it can be hard to factor out therapist effects, as can be seen in this report.

For a qualitative study like this one, the Introduction and literature review should lead to one or more 'Research Questions' that are stated at the end of the Introduction. These are 'how' and/or 'why' questions rather than 'what' or 'whether' questions. In turn, the Research Questions determine the choice of methodology.

Additional reference to the existing, limited literature base have been added to the background. We have also added additional references to the issues of arts-based interventions in the final section of the introduction, and the acknowledgement that this part of a pilot cluster RCT.

As this was a qualitative process evaluation the research question is focused on ‘whether’ the intervention is acceptable, and why or why not. I have altered the final sentence of the introduction to better reflect the aim of the process evaluation, as was stated in a previously published protocol (which is cited in the paper)

The greatest shortcoming of this paper is that intervention is not described! It needs to be described in detail, right after the section on recruitment. We need to know what the patients experienced: what did they do, how many sessions did they have, how long were the sessions...? How was the intervention derived? Does it mirror interventions used previously in similar settings? Information is needed about the person who provided the intervention. Did she have training or previous experience with this kind of intervention? Does she have art therapy training (North American art therapists reading the paper will want to know this)? This information is important for understanding how the intervention might be used in the future, in other settings.

Some people in the art therapy world might argue that art therapy training is necessary for delivering this kind of intervention. I don't feel that way myself. Rather, I am interested in finding ways for health professionals who are not art therapists to deliver arts-based interventions so arts-based interventions can be more widely available. Toward this end it is important to discuss the skills/credentials/experience of the facilitator and perhaps make suggestions about the qualifications needed.

We have added in a paragraph detailing the intervention, this should give an idea of the focus of the sessions. The arts-based intervention was not art therapy, meaning that the participants did not have to base the art around their own mental health or emotions. Participants who did engage in creative writing could choose their own prompts or stories that they wanted to tell, while this obviously related to personal experience or related to subjects of great personal significance, the focus was not on content or output but on the process of learning the skill of creative writing or drawing. The distinction between art therapy and arts-based intervention is now briefly described in the intervention section.

The fact that there was only one facilitator has now been highlighted as a limitation in the discussion. We have also provided a brief description of the facilitator in the section about the intervention.

If any patients dropped out, this needs to mentioned, with reasons given.

No patients dropped out of the intervention. Some patients did drop out of the pilot cluster RCT but this was not explored in the process evaluation. This is described in the pilot cluster RCT results in a different publication (cited in the manuscript).

The intervention included creative expression (painting, drawing and writing?) and one-on-one facilitation. These two components of the intervention might have been equally important, or one might have been more important than the other. If the authors have data that could shed light on this question it would be helpful to add that information in.

There is no other data available on this within the evaluation, as all components had been introduced through a development process (development paper is now referenced in text), and all aspects were seen as important . One-to-one facilitation was discussed in terms of acceptability of delivery, and it was seen as a beneficial aspect (described in the second theme), while creative expression was not identified as a sub-theme relating to the benefits of the intervention. However it wouldn’t be appropriate to label one as more important than the other considering the nature of pilot trials and the fact we were not assessing for effectiveness.

Was it a wait-list control group?

Yes, this has been clarified in the text now. However the interviews took place prior to the control group receiving any sessions of the intervention as the wait list could only start once the full study had closed.

Results: these are well presented, but missing is a discussion of comments that did not fit with the themes. There probably were some things that didn't go well, or some patients who didn't want to continue, or something along these lines. Negative comments, or exceptions to the main themes, need to be mentioned for the presentation of the results to be complete.

The only negative comments from the interviews related to the length of the intervention, which is described in theme three. The results were very consistent across all interviews.

Discussion: A qualitative study cannot "conclusively establish an effect" and maybe the words "effect" and "impact" are used too much in the Discussion. Effects and impacts were mentioned by the interviewees, but they were not demonstrated, established or proven. The focus needs to stay on the potential mechanisms that the qualitative analysis revealed.

I have removed the term impact and effect where it is inappropriate, I also think this point stands for the naming of a number of themes/subthemes so I have changed the phrasing of those as well.

The Discussion overall is very good. 

A big problem with this kind of study (novel arts-based intervention with only one facilitator) is that it is impossible for readers to assess the facilitator effect. Maybe the facilitator was so nice to be with that anything she did with the patients would have lead to comments about beneficial effects! Maybe she was an outstanding facilitator and a similar intervention delivered by a different person would not have seemed acceptable to patients. This problem needs to be addressed and included as a limitation.

A larger study on this topic, whether an RCT or some other type of study, would need to have more than one facilitator. 

The issue of the single facilitator has been highlighted in the limitations section of the discussion now, and a description of the facilitator has also been provided in the intervention summary.

We completely agree that any future study would require more than one facilitator in order to assess the true effect and also fidelity of delivery.

One of the reasons I think this paper is important is that arts-based interventions can have powerful positive effects in healthcare settings at relatively low cost. The main question in my mind after reading the paper was: who can facilitate this intervention for ESKD patients? Can it be a nurse who is there anyway, perhaps a nurse with some experience with the arts? Does it need to be someone else? If so, who pays for that? Etc.

We really appreciate you raising this point as it is one of the key findings from this study. We have now addressed this issue in the final paragraph of the discussion, and link it to recent work we have been doing regarding using volunteers to deliver this type of intervention in a more sustainable manner. As the person who delivered this was not a renal nurse, or a professional artist, the intervention highlighted that with some training people with good interpersonal skills and an enthusiasm and understanding of arts, can facilitate these interventions without needing specialised clinical skills or education in the arts.

Congratulations on an excellent paper.

Thank you very much - And thank you for such constructive comments!

Reviewer 3 Report

I recommend at a minimum the inclusion of more detail about:

*prior research supporting art therapies as an intervention in the populations, 

*more detail about the pilot study done before this study

*a description of the patient population with demographic and other medical data

  • an explanation as to why this is being called a randomized controlled trial.

To greatly enhance this article, I would recommend looking more closely at the theoretical framework, highlighting its usefulness in nursing, and describing how the intervention hoped to achieve this aim, and where it failed or only moderately met the aim. of course, suggestions to improve on achieving the results consistent with the theoretical framework of flow would be needed also. I think this would be the most valuable route to enhance this presentation and would then add a significant and innovative manuscript of interest to nurses and others. 

Author Response

Thank you so much for taking the time to review this manuscript and provide constructive comments.

I recommend at a minimum the inclusion of more detail about:

prior research supporting art therapies as an intervention in the populations, 

Additional reference to the existing, limited literature base have been added to the background. We have also added additional references to the issues of arts-based interventions in the final section of the introduction, and the acknowledgement that this part of a pilot cluster RCT. This article is also not looking at art therapy, but arts-based interventions, and the distinction has now been made in the description of the intervention that has been added

more detail about the pilot study done before this study

This is a process evaluation of a pilot cluster RCT. The protocol for this has been previously published, as have the results of the pilot trial. This is the qualitative portion of the study that specifically relates to the acceptability of the intervention. This has been clarified now in the design section of the manuscript.

a description of the patient population with demographic and other medical data

Additional detail describing participants has been added, including the recruitment section and the participants section.

an explanation as to why this is being called a randomized controlled trial.

This is a process evaluation of a pilot cluster RCT. The protocol for this has been previously published, as have the results of the pilot trial. This is the qualitative portion of the study that specifically relates to the intervention. This has been clarified now in the design section of the manuscript.

To greatly enhance this article, I would recommend looking more closely at the theoretical framework, highlighting its usefulness in nursing, and describing how the intervention hoped to achieve this aim, and where it failed or only moderately met the aim. of course, suggestions to improve on achieving the results consistent with the theoretical framework of flow would be needed also. I think this would be the most valuable route to enhance this presentation and would then add a significant and innovative manuscript of interest to nurses and others. 

This has been added to the introduction, with an overview of the theoretical framework of flow and how this connects to improvements in mental health and wellbeing. Additionally a description of the intervention, including how it relates to the theoretical framework, has been added in a new section describing the intervention in more detail.